# Integration of Task-Based Exoskeleton with an Assist-as-Needed Algorithm for Patient-Centered Elbow Rehabilitation

**DOI:** 10.3390/s23052460

**Published:** 2023-02-23

**Authors:** Pablo Delgado, Yimesker Yihun

**Affiliations:** Department of Mechanical Engineering, Wichita State University, Wichita, KS 67260, USA

**Keywords:** assist-as-needed, exoskeleton, robot-therapy, rehabilitation

## Abstract

This research presents an Assist-as-Needed (AAN) Algorithm for controlling a bio-inspired exoskeleton, specifically designed to aid in elbow-rehabilitation exercises. The algorithm is based on a Force Sensitive Resistor (FSR) Sensor and utilizes machine-learning algorithms that are personalized to each patient, allowing them to complete the exercise by themselves whenever possible. The system was tested on five participants, including four with Spinal Cord Injury and one with Duchenne Muscular Dystrophy, with an accuracy of 91.22%. In addition to monitoring the elbow range of motion, the system uses Electromyography signals from the biceps to provide patients with real-time feedback on their progress, which can serve as a motivator to complete the therapy sessions. The study has two main contributions: (1) providing patients with real-time, visual feedback on their progress by combining range of motion and FSR data to quantify disability levels, and (2) developing an assist-as-needed algorithm for rehabilitative support of robotic/exoskeleton devices.

## 1. Introduction

As people get older, they may experience age-related neuromuscular and sensorimotor degeneration, resulting in disabilities and limiting the range of motion (ROM) of joints and coordination. With an aging population, healthcare systems will face significant challenges to meet the growing demand. The COVID-19 pandemic has highlighted the need for safe and effective home-based and telerehabilitation settings. Cost-effective automation devices like exoskeletons, equipped with safety and assist-as-needed controls, are critical for the success of these interventions and rehabilitation programs. Home-based and telerehabilitation programs can reach more patients while reducing physician interactions and overall rehabilitation costs [1,2].

Individuals who have lost their range of motion (ROM) for different reasons are commonly advised to pursue physical therapy as a means of recovering their lost ROM [3,4,5]. Physical therapy can be a demanding, lengthy, and costly process, which can cause individuals undergoing therapy to lose interest. One potential solution to motivate individuals with disabilities to participate and continue physical therapy is to reduce recovery time and provide real-time feedback to promote motivation. By implementing these strategies, the cost of treatment can also be reduced. Research has shown that high-intensity repetitive tasks can improve recovery time for patients undergoing physical therapy [6]. Increasing the number of repetitions and intensity of an exercise during a therapy session may impact its quality, as it can lead to physical therapist fatigue, particularly if multiple patients are treated within a short time frame. To mitigate this challenge, incorporating robotic devices into the therapy treatment can provide a more efficient approach [7]. Most of these robots, exoskeletons, have different sensors that can be used to collect data for further analysis, such as for safety and to give feedback to patients regarding the treatment and their progress [8].

Aside from the choice of exoskeleton type (joint-based [9,10] or task-based [11,12]), the control algorithms to drive them are an important factor that can help to improve the outcome of physical rehabilitation treatments. A physical therapist assists the patient depending on the severity of the joint impairment. As the treatment progresses, the patient may need less assistance from the professional caregiver, to the point where the patient does not need assistance at all [13]. Researchers have recently focused on assist-as-needed (AAN) algorithms as a means of enabling exoskeletons to detect when assistance is necessary for a person to perform a given task. Various AAN algorithms have been proposed in the literature, including impedance control. This control strategy aims to establish a connection between the force applied and the trajectories followed, modeled as a mass-spring-damper system. In the field of rehabilitation, this AAN approach has been utilized by estimating the algorithm parameters specific to each individual patient. In the study presented in [14], a three-degree-of-freedom (DOF) mechanism was utilized, which was constrained to a planar circular motion. However, it should be noted that this control strategy can only adjust the torque supplied by the exoskeleton, as it assumes accurate knowledge of the torques/forces generated by the human joints. Furthermore, the mass, spring, and damper parameters must be identified for each subject individually. An alternative AAN method documented in the literature is to estimate joint torques by analyzing surface electromyography (sEMG) signals that are produced by the joint muscles, as demonstrated by George et al. [15]. Hu et al. [16] also employed this technique to design an assist-as-needed (AAN) algorithm for controlling an elastic cable-driven elbow flexion-extension exoskeleton. They estimated joint torques offline, which differs from impedance control, as it estimates external joint torques/forces provided by the subject. However, this approach only relies on sEMG information, which can vary over time, leading to a decline in controller performance, particularly in patients who are commencing physical therapy. Although these AAN algorithms offer assistance to patients, they require patients to be able to move their limbs partially within the range of motion (ROM) of the given task to undergo training.

In this study, we used a task-based exoskeleton [17] designed to replicate the elbow’s ROM, which is capable of providing assistance to patients going through physical therapy treatments. In addition, an AAN scheme that is capable of providing support to the patient as long as it is needed for an elbow flexion and extension exercise is proposed. Instead of focusing on building a model to predict the torques generated by the patients’ joints, we build a model to predict when the patient is moving their limbs by themselves utilizing low-cost FSR sensors, allowing the exoskeleton to act as a follower without providing support, especially to patients that are starting the physical-rehabilitation treatment, where the use of physiological signals to establish a relation between the human-joint torques/forces is not a suitable strategy. On the other hand, if the patient is not able to move the elbow joint, the algorithm will allow the exoskeleton to provide assistance and allow the wearer to finish the task. To assess the mechanism and AAN algorithm’s efficacy, a human-subject test was conducted on five people with disabilities. The sEMGs from the main flexor muscle, the bicep, were measured and analyzed to determine when the individual was flexing their arm independently and when the exoskeleton was providing complete support.

The remainder of this paper is structured as follows: Section 2 describes the mechanism synthesis procedure and the resulting mechanism. In Section 3, the 3D-printed prototype of the exoskeleton, the coupled human-exoskeleton model, and the hardware used in the study are presented. Section 4 outlines the proposed AAN algorithm. The real experiments and their findings are presented in Section 5. Lastly, the paper concludes with Section 6, which includes the study’s conclusions and recommendations for future research.

## 2. Mechanism Synthesis

This study uses a Bennet Linkage [18], a spatial 1-DOF parallel mechanism with four revolute joints, as a task-based exoskeleton to create elbow flexion and extension motion. Most researchers have modeled the elbow joint as a simple hinge joint, assuming the elbow joint axis is fixed in its range of motion [19,20,21]. However, research conducted with electromagnetic sensors has exposed that the joint’s axis moves throughout its range of motion, leading to three-dimensional motion of the forearm [22]. In this study [22], Bottlang et al. modeled the ulna rotation around the humerus using screw displacement axes, and the results showed that these displacements varied from 2.6–5.7${}^{\circ }$ and 1.4–2.0 mm in translation. similarly, comparable results were found through simulation in our previous work [23], where an assessment of a joint-based exoskeleton was performed in the musculoskeletal software, OpenSim. Thus, rather than utilizing a simplified 1-DOF hinge joint (joint-based) to simulate the elbow joint, this study has devised a task-based 1-DOF special mechanism to generate elbow flexion-extension motion.

The synthesis was performed based on spatial kinematic information from a state-of-the-art motion capture system while a person is executing the task. Based on the trajectories generated from elbow flexion–extension, it was determined that a Bennet Linkage is a suitable solution to replicate the motion. Once the topology of the task-based exoskeleton is chosen, then the position and orientation of the trajectory points are transformed into dual quaternions. these are then utilized in an optimization algorithm to determine the link dimensions and joint orientations for the Bennet Linkage while meeting the mechanism constraints. The flowchart of the synthesis procedure is presented in Figure 1. The results of the approach in our work were used to model and build a prototype in computer-aided design (CAD) software. The resultant mechanism is shown in Figure 2. The detailed synthesis procedure is discussed in our previous work [17].

## 3. Exoskeleton Prototype and Materials

The exoskeleton prototype obtained through the synthesis procedure discussed in Section 2 is presented in Figure 3. The spatial four-bar mechanism has been 3D printed using PLA material, and it is equipped with metal reinforcement. Its active joint is powered by a NEMA 23 Stepper Motor with a 46:1 Planetary Gearbox, Figure 4b. The NEMA 23 stepper motor is driven by a control unit that contains a Teensy 4.1 development board with a Cortex-M7 as a processor unit, a micro-stepper driver, and a 24-volt power supply, see Figure 4a. Additionally, the task-based exoskeleton is equipped with passive joints that can be adjusted to different subjects’ anthropometric measurements, these are highlighted in red in Figure 3. In addition, the exoskeleton has two attachments to secure the wearer’s arm and forearm. One attachment confines the wearer’s arm to the ground, while the other constrains the forearm to move along with the exoskeleton across its range of motion.

To acquire feedback from the wearer while the exoskeleton is active, two different types of sensors are used: a Force Sensing Resistor (FSR) [24], see Figure 5, and surface electromyography (sEMG) sensors, Delsys Research system, see Figure 6. The FSR is utilized to measure the force exerted by the wearer on the forearm holder during task execution. The data from the FSR is integrated into the proposed AAN control strategy in this study. The FSR sensor is mounted onto the forearm holder as shown in Figure 3. The application of force on the FSR results in a decrease in its electrical resistance. This change in resistance, combine with a 10 KΩ and 5-volt power supply, leads to a variation of the voltage drop at the FSR terminals. This variation in voltage is measured by a 10-bit analog–digital converter peripheral on an Arduino MEGA 2560 board, see Figure 5.

On the other hand, the sEMG sensors, placed on the bicep and tricep muscles, provide information on the level of muscle engagement in millivolts. This information can be used to assess the impact of the exoskeleton on the wearer. The Delsys acquisition system possesses 16 channels and is capable of collecting data at a sampling rate of approximately 2000 KHz. This system comes with the sEMG Works software, which allows the user to collect and analyze sEMG data by setting up timed experimental tasks. Additionally, the manufacturer provides the SDK (Software Development Kit) and API (Application Programming Interface) that allow the integration of the Delsys Acquisition system with 3rd-party software. For our purposes, we used the API in order to synchronize the sEMG data collection process with the FSR sensor, and the estimated position of the stepper motor, θ(t).

The interface schematic of the human-exoskeleton system with the sensors is presented in Figure 7. The position of the motor is estimated based on the number of steps it takes, while the FSR measurements are sampled at a frequency of 100 Hz. This information is sent to the main computer, using the RS232 communication protocol, where the high-level control logic is processed. Likewise, the sEMG measurements are sent to the computer through USB communication. In the computer, a Python interface has been developed in order to communicate with each one of the peripherals and process the information received. This is done to create a mathematical model to infer when the wearer requires assistance while using the task-based exoskeleton.

## 4. Proposed Assist-as-Needed Algorithm

We propose an Assist-as-Needed (AAN) algorithm to identify instances where the wearer cannot execute the elbow-flexion task using FSR measurements and feedback on the input angles, θ(t), on the exoskeleton. The information obtained is used to develop a mathematical model to infer the period of inactivity of the person by employing the supervised machine learning algorithm, Least-Squares Regression Methods [25].

### 4.1. Model Development

A preliminary analysis was conducted to study the FSR measurements while a person rested their forearm, as shown in Figure 3, on the exoskeleton during 10 repetitions of changes in input angle from 0${}^{\circ }$–60${}^{\circ }$ and vice versa. To reduce external noises affecting the FSR measurements, a 2nd-order Butterworth filter at a sample rate of 100 Hz was implemented. The point-cloud distribution of the FSR changes is shown as a scatter plot in Figure 8. As can be observed, the FSR measurements at different angle values are not constant during the whole experiment.

A weighted $L^{2}\text{-}norm$, Equation (Equation 1), was used to create another representation of the distribution of the FSR point cloud. The norm takes into account both the exoskeleton input angle, θ(t), and the FSR measurement values. The resulting plot is shown in Figure 9.
(1)$$z=\sqrt{\theta^{2}\left(t\right)+{\left(w\times FSR\left(t\right)\right)}^{2}}$$

The FSR measurements range from 0 to 5 V, while the angle information ranges from 0 to 60${}^{\circ }$. In this study, the measurements of both FSR and angle are important. Therefore, a distinctive characteristic is computed based on the $L^{2}$-*norm* of these two measurements. Additionally, considering the different ranges and relative importance of both values in the model, a weighting factor, *w*, is applied to the FSR measurements to increase their significance and impact in the predicting model. In our experiment, this value was set to 15, i.e., w=15. A mathematical model to establish each individual’s resting area based on the distribution shown in Figure 9 could be developed to detect when subjects flex their elbow unassisted. Subjects that flex their arm by themselves will apply less pressure onto the FSR sensor, leading to low measurements from the sensor, resulting in a decrease in the weighted *l*${}^{2}$-*norm* calculated using Equation (Equation 1). The opposite can be detected as well since a higher FSR measurement will indicate that the wearer is applying more force to the FSR sensor. These different scenarios will yield three different regions, indicating whether the subjects are flexing, extending, or resting their arm.

Our goal in this section is to determine a mathematical model that can infer the intention of the wearer using these 3 different regions and drive the exoskeleton accordingly. For this purpose, least-square regression methods are used. To describe the behavior of the resting region, Figure 9, a curve representing the mean of the distribution across different input angles must be determined. Then, lower and upper boundary lines enclosing the resting area are defined by 2 times the standard deviation (std), σ, away from the mean curve. This process is repeated across *n* different segments of the distribution, see Figure 10.

The mean curve of the distribution is determined by approximating each segment to a linear line, ${\hat{z}}_{i}=a_{i}\times \theta \left(t\right)+b_{i}$. The parameters of the line are determined by minimizing the residual sum of squares (RSS), Equation (Equation 2), between the training data and the predicted value, least-square linear regression.
(2)$$RSS\left(a_{i},b_{i}\right)=\sum_{k=1}^{n}{\left(z_{k}-\left(a_{i}\times \theta_{k}+b_{i}\right)\right)}^{2}$$
where *i* represents the segment, *n* is the number of data points in segment *i*, and $a_{i}$, and $b_{i}$ are the coefficients representing each ${\hat{z}}_{i}$ line. This problem can be solved by grouping the lines in matrices, as presented in Equation (Equation 3).
(3)$$\underset{Z_{i}}{\underset{︸}{\left[\begin{array}{c} z_{1}\\ z_{2}\\ \vdots \\ z_{n} \end{array}\right]}}=\underset{A_{i}}{\underset{︸}{\left[\begin{array}{cc} \theta_{1} & 1\\ \theta_{2} & 1\\ \vdots & \vdots \\ \theta_{n} & 1 \end{array}\right]}}\underset{X_{i}}{\underset{︸}{\left[\begin{array}{c} a_{i}\\ b_{i} \end{array}\right]}}$$

Then, the least-square solution to Equation (Equation 3) can be solved by Equation (Equation 4).
(4)$$X_{i}={\left(A_{i}^{T}A_{i}\right)}^{-1}A_{i}^{T}Z_{i}$$

Once the coefficients of each of the lines are determined, the standard deviation of each region is found, $\sigma_{i}$, to bind the data points to $\pm 2\times \sigma_{i}$. The boundaries of the segment of the distribution must enclose 95% of the data since each segment distribution resembles a normal distribution [26]. The output of this procedure is presented in Figure 11. where the red curve with squared markers represents the mean line of each segment, ${\hat{z}}_{i}$, the blue-dot points represent the cloud-point distribution of the data, and the green lines with dot markers represent the linear boundaries found for each segment.

The information obtained from the boundary lines of Figure 11 is used to determine the upper and lower boundary curves that surround the cloud-point distribution representing the resting region. Each boundary line is approximated to a 2^*nd*^-order polynomial, Equation (Equation 5):(5)$$m\theta^{2}\left(t\right)+n\theta \left(t\right)+o$$
where the *m*, *n*, and *o* are the coefficients to be found for each one of the curves. This problem is similar to the one shown in Equation (Equation 3), by modifying matrix *A* to include the coefficient of $\theta^{2}\left(t\right)$ as shown below by Equation (Equation 6):(6)$$\underset{Z}{\underset{︸}{\left[\begin{array}{c} {\hat{z}}_{1}\\ {\hat{z}}_{2}\\ \vdots \\ {\hat{z}}_{n} \end{array}\right]}}=\underset{A}{\underset{︸}{\left[\begin{array}{ccc} \theta_{1}^{2} & \theta_{1} & 1\\ \theta_{2}^{2} & \theta_{2} & 1\\ \theta_{3}^{2} & \vdots & \vdots \\ \theta_{n}^{2} & \theta_{n} & 1 \end{array}\right]}}\underset{X}{\underset{︸}{\left[\begin{array}{c} m\\ n\\ o \end{array}\right]}}$$

Now, this problem can be solved as expressed in Equation (Equation 4), obtaining coefficients for the upper boundary and lower boundary curves, respectively. An example of the resultant boundary curves is presented in Figure 12, where the green curve with square markers is the lower boundary, the magenta curve with dot markers is the upper boundary, and the blue-dot points represent the data used for training this model, the resting region. Then, the information extracted from the FSR measurement and the exoskeleton input angles, θ(t), can be classified into 3 different regions. Any point that lies in the first region, region 1, can be interpreted as the user trying to flex the elbow. Similarly, any point in the second region, region 2, is classified as the person resting the arm on the exoskeleton without applying any effort/force. Lastly, any point above the upper boundary curve is classified as the person exerting force against the exoskeleton, region 3. This can be interpreted as the person attempting to extend the forearm.

The above procedure can be executed for each individual to obtain a unique model for them during the rehabilitation session. The algorithm summarizing the training process, Algorithm 1, is presented below:
**Algorithm 1:** Mathematical model training.**Require**:
Training Data Set FSR and θ**Ensure**:
Trained Model, LUC and LLC1:w←w                 ▹*w* is the weight for FSR measurements2:K←K                  ▹*K* Number of segments to split data3:$z=\sqrt{\theta^{2}+{\left(w\times FSR\right)}^{2}}$4:Split *z* in *k* Segments5:**for**i=1:K**do**6:    $A_{i}=\left[\theta_{i},\;1^{n\times 1}\right]$7:    $X_{i}={\left(A_{i}^{T}A_{i}\right)}^{-1}A_{i}Z_{i}$8:    ${\hat{Z}}_{i}=\theta_{i}X{\left(1\right)}_{i}+X{\left(2\right)}_{i}$9:    $e={\hat{Z}}_{i}-Z_{i}$10:    $\sigma_{i}=std\left(e\right)$                  ▹ Get Standard Deviation11:    $LU_{i}={\hat{Z}}_{i}+2\times \sigma_{i}$                    ▹ Upper Line12:    $LL_{i}={\hat{Z}}_{i}-2\times \sigma_{i}$                    ▹ Lower Line13:**end for**14:$Y=[LU_{1},\;\cdots ,\;LU_{K}]$15:$A=[\theta_{i}^{2},\;\theta_{i},\;1^{n\times 1}]$16:Upper Boundary Curve, $LUC={\left(A^{T}A\right)}^{-1}A\;Y$17:$Y=[LL_{1},\;\cdots ,\;LL_{K}]$18:Lower Boundary Curve, $LLC={\left(A^{T}A\right)}^{-1}A\;Y$19:Save curve Model: LLC and LUC

### 4.2. Classification and Assist-as-Needed Strategy

To detect the intended wearer’s action, the obtained model defined by the curves LUC and LLC is used to determine in which region of the trained model the current weighted $l^{2}$-norm value, Equation (Equation 1), lies with respect to the current input angle θ. For this purpose, a classification algorithm is presented in Algorithm 2. In this algorithm, the weighted $l^{2}$-norm value, *z*, is compared with respect to two predicted values. ${\hat{z}}_{u}$ and ${\hat{z}}_{l}$, which correspond to values on the upper boundary curve UBC and lower boundary curve LUC, respectively. If the difference between *z* and ${\hat{z}}_{u}$ is greater than 0, then the point is above LUC, Region 3. On the other hand, if the difference between *z* and ${\hat{z}}_{l}$ is less than 0, then the point lies in Region 1. However, if none of the previous conditions hold, then the point is on Region 2.

Detecting the wearer’s action allows the exoskeleton to be commanded to perform different actions depending on its state. If the intention of the wearer is to perform an elbow flexion (region 1), then, the exoskeleton should follow the wearer through the range of motion. Similarly, the same can be done when the wearer is extending the forearm (region 3). However, if the wearer is not capable of performing the action by themselves, then the exoskeleton should assist them through the rest of the range of motion. To determine when the wearer needs assistance, the exoskeleton will be programmed to detect the time spent on the resting region (region 2). Once the exoskeleton detects that the user has spent too much time in the resting zone, the exoskeleton will take over and finish the exercise for them. The novelty of this approach lies in its adaptability to the regions of interest for each individual using the exoskeleton and its low-cost implementation.
**Algorithm 2:** Classification algorithm.**Require**:
Trained Model Curves LUC and LLC, and current FSR and θ values**Ensure**:
Detected Action1:w←w                     ▹*w* is the weight for FSR measurements2:$z=\sqrt{\theta^{2}+{\left(w\times FSR\right)}^{2}}$3:${\hat{z}}_{u}=LUC\left(1\right)\;\theta^{2}+LUC\left(2\right)\;\theta +LUC\left(3\right)$4:${\hat{z}}_{l}=LLC\left(1\right)\;\theta^{2}+LLC\left(2\right)\;\theta +LLC\left(3\right)$5:**if**$z-{\hat{z}}_{u}>0$) **then**6:    Return: Action = Extension                         ▹ Region 37:**else if**$z-{\hat{z}}_{l}<0$**then**8:    Return: Action = Flexion                          ▹ Region 19:**else**10:    Return: Action = No action                          ▹ Region 211:**end if**

## 5. Experimental Testing

### 5.1. Participant Description

In order to showcase the effectiveness of the proposed AAN algorithm presented in Section 4, an experiment with different subjects wearing the exoskeleton was performed. The selected task involved performing elbow flexion from $0^{\circ }$ to $60^{\circ }$ 20 times. The first 10 trials were used to train the proposed AAN. Then, the remaining trials were performed by either the person or the exoskeleton, depending on whether the subject could finish the task. The human-subject testing was performed in collaboration with the Cerebral Palsy Research Foundation (CPRF), where five subjects volunteered to be part of the experiment. The testing group was composed of one female and four males, with an age that varies from 23 to 63 years old. Four out of the five subjects had different levels of spinal cord injury (SCI) [27], and one had Duchenne muscular dystrophy (DMD) [28]. The protocol of the study as well as the results of the experiments are presented in the following subsections. The AAN algorithm would only be activated to help the participants to complete the elbow-flexion task when it detected the user could no longer flex the forearm. In addition, sEMG sensors were used to measure the bicep activity during the elbow-flexion task, to give some feedback on the effort applied at the end of the session. In addition, IRB approval was obtained to proceed with the experiment from the Institutional Review Board Committee at Wichita State University.

### 5.2. Protocol

Prior to commencing the experiment, the experimental procedure was comprehensively explained to all participants, and each individual diligently signed the requisite IRB forms. The following step consisted of having two sEMG sensors placed on the participant’s arm, post cleansing the skin using alcohol pads. These sensors were placed specifically on the bicep and tricep muscles. These are the main flexor and extensor of the elbow, respectively. Then, the subjects were asked to sit next to the exoskeleton in order to adjust it to the participant’s anthropometric measurements. The CPRF personnel were responsible for ensuring that the exoskeleton was appropriately aligned and fitted. All necessary adjustments and alignments were accomplished by manipulating the passive joints shown in Figure 3. Once the adjustments were made, the participant’s arm was attached with Velcro to the arm and forearm holders, respectively (Figure 13). The compliant straps were used to accommodate different-sized patients and to constrain unwanted movements at the attachment points. after the patient was attached to the mechanism, they were physically constrained to follow the spatial displacement of the Bennet linkage without being influenced or propelled by the straps. In this stage, the subject was instructed to relax and rest their arm while the exoskeleton executed ten elbow flexion and extension motions at an angular velocity of 20 degrees/s from 0 to 60${}^{\circ }$. This range of motion was set between 0–60${}^{\circ }$ to ensure the safety and maintain consistency in the experimental setup across the subjects since they were wheelchair bound. Upon completion, the AAN algorithm model was trained, and the participant was asked to attempt the remaining number of flexion repetitions. In the event that the participant was incapable of concluding one or more of the repetitions, the exoskeleton would have taken over and accomplished the task on their behalf. Finally, upon completion of the task, the wearer’s arm and forearm were released from the exoskeleton, by carefully removing the Velcro from both places. In addition, the sEMG sensors were also removed.

During the experiment, the exoskeleton was controlled by a Python-based Graphical User Interface (GUI), see Figure 14. In this GUI, the range of motion (in degrees), the speed in (degrees/second), and the number of trials to be performed could be specified. In addition, whether sEMG sensors were used and whether the AAN was activated could have also been selected. Furthermore, the exoskeleton could be instantaneously stopped in the event of an emergency by pressing the Stop button.

### 5.3. Results

The experimental results obtained from each one of the subjects are presented in this section. The accuracy of each of the models is presented in Table 1. The accuracy of the model was computed by determining the number of predictor values $Z_{i}$ that were classified correctly, TP, divided by the total number of samples utilized in the training section, N, multiply by 100%, as depicted in Equation (Equation 7). The average model prediction accuracy across participants is 91.225%. These models were fed up to Algorithm 2 to determine each subject’s intention through the rest of the task. During the sessions described in the protocol, the sEMG values of the bicep muscle were recorded to analyze its Root Mean Square (RMS) and provide feedback on the amount of effort applied by the patient at the conclusion of the session.
(7)$$model\;accuracy=\frac{TP}{N}\times 100\%$$

In Figure 15, the model to predict the action of subject one is presented along with the feature points extracted from the FSR sensor and the input angles of the stepper motor for the 10 trials outside the training session. This subject has SCI and has been diagnosed with general atrophy and a limited active range of motion and strength for all joints. The figure shows that the subject was capable of performing the requested task; however, as the exercise progressed, the subject started to experience some difficulties. These moments are highlighted by a black circle on top of the Elbow Curl Angle graph since the exoskeleton stops due to lack of activity from the subject to analyze whether it needs to finish the task for them. Moreover, to demonstrate the correlation of the FSR measurement to the algorithm output, the FSR measurements were added for the subject, see Figure 15.

Similar results were observed from subject 2, Figure 16, who has a higher limitation in their range of motion. This was noticeable in their results since in eight out of the ten trials, the subject was unable to complete the task, and the exoskeleton had to complete it.

Subject 3, Figure 17, presented some difficulties in the first two trials. However, the patient was able to finish the rest of the trials by themselves. Unlike the previous two subjects, this patient has been diagnosed with an incomplete SCI. Therefore, the difficulties observed at the beginning may be attributed to the subject getting familiar with the exoskeleton.

On the other hand, subject 4, Figure 18, was unable to complete any of the trials. The feature point never left region 2 from the predicted model. This subject, in contrast to the previous ones, has Duchenne muscular dystrophy and has been diagnosed with greatly reduced strength and active range of motion in all joints. As can be observed from the figure, the exoskeleton waited an appropriate amount of time to detect any effort from the subject. Lastly, subject 5, Figure 19, presented similar results as subject 3. This patient has also been diagnosed with an incomplete SCI.

From all subjects, it can be observed that the proposed Algorithm 1 from Section 4 segmented the subject’s intention during the elbow flexion task into three different regions. This was achieved by creating a subject-based predictive model for each individual.

The AAN control strategy presented was capable of adapting to each individual with a minimum number of trials, as demonstrated by the FSR sensor measurement. The time required to train the model for each individual was 1 min. Unlike previous works, the proposed AAN does not rely on the joint forces/torques from the subject as feedback for the control algorithm, such as the impedance AAN control scheme presented in [14]. In addition, the time and complexity to calculate the impedance parameters to adjust the AAN to each individual would make it unfeasible for clinical trials. On the other hand, an AAN algorithm that depends only on sEMG information to estimate the joint torques using a neural network with a 97% of accuracy has been presented in [29]. However, the authors concluded that due to the number of parameters needed (137), it is very hard to deploy this strategy in a high-frequency real-time experiment. In addition, the AAN strategy was only implemented in subjects with no disabilities. In our case, the proposed approach has been implemented in people with disability.

### 5.4. sEMG Analysis Results

The sEMG values of the Bicep muscle of each individual subject were recorded during the experiments. The objective of this signal was to offer patients feedback on the level of effort they exerted during the exercise. In the literature, it is very common to use the normalized windowed Root Mean Square Value (RMS) to estimate the effort provided by a muscle [30]. For normalizing the sEMG, usually, the Maximum Voluntary Contraction (MVC) for each muscle being studied is used. MVCs values are performed using static loads to obtain a maximum contraction of the muscle. However, it is not advisable to obtain MVCs when working with patients undergoing physical therapy since they may not be able to perform them, and it may not be safe for them to do so. In addition, due to muscle impairment, the subject may suffer from sensitive tissues, which would cause them pain. In this case, another clinical term could be used to normalize the RMS values of the sEMG signals, Acceptable Muscle Contraction (AMC). The AMC in this study will be taken as the maximum sEMG value presented during the rehabilitation session.

The effort of the bicep muscle for each one of the subjects is presented in Figure 20, Figure 21, Figure 22, Figure 23 and Figure 24. In these figures, a bar plot is used to show the RMS value of each trial sEMG, and a red-colored line is used to indicate the mean value for the RMS of the sEMG signals during the training trials. Subject 1, presented a mean effort for the training session of around 34%, then, the bicep effort started increasing as the subject was applying some torque at the joint to perform the requested elbow-flexion task. The same can be said for the rest of the patients; the only difference among them was the mean sEMG values computed during the training session. One peculiar case that we found in our study is that subject 4 could not perform the task by themselves; however, the patient was able to engage the bicep muscle to produce effort, Figure 23. Therefore, this information could be used to inform the patient of the endurance performance of the bicep during the exercise. This information will keep them motivated to continue the physical-therapy treatment regardless of whether they can finish the exercise or not.

## 6. Conclusions

In the presented work, an AAN strategy alongside a task-based exoskeleton was presented to help people going through physical therapy. The proposed strategy was tested with five individuals, all of them have a disability. The algorithm was able to adapt to each individual by creating model profiles that were used to know when they were not capable of performing the elbow-flexion task. The average of the models’ accuracy was 91.22% with an acceptable number of trials that could be used as the warm-up session for the task since the AAN algorithm does not depend on bio-electrical signals as other algorithms do. In addition, sEMG signals of the bicep were used to obtain feedback from the therapy session, showing valuable information that could be used by the physical therapist to provide feedback to the patients. Keeping records of the progress that the patients are making is an important process of physical therapy for them since, psychologically, it helps them to remain motivated to keep assisting in the sessions. In summary, the contributions of this paper are the following: first, a subject-centered AAN algorithm was developed for rehabilitative treatments, and second, visual feedback in the form of effort using sEMG is given to patients to keep track of their progress. In future works, we are planning to expand the current work by adding performance factors based on the collected sEMG signals. Moreover, this information could be used to detect when the patient is slacking off and has become technologically dependent on the exoskeleton. Additionally, a virtual reality environment will be explored as a means to keep the patients motivated during the therapy session, by adding games that simulate the task being performed with the exoskeleton.

## Figures and Tables

**Figure 1 sensors-23-02460-f001:**
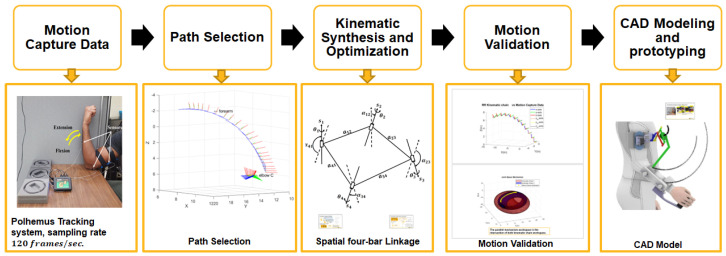
Exoskeleton synthesis methodology.

**Figure 2 sensors-23-02460-f002:**
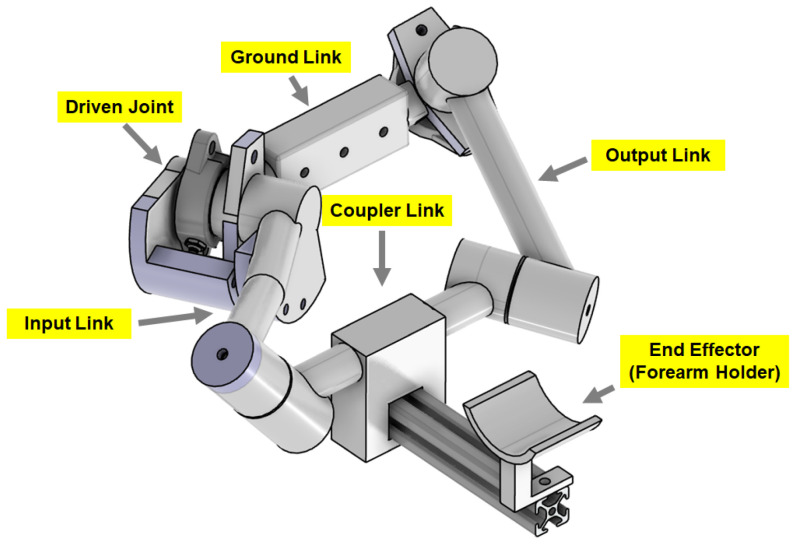
Exoskeleton CAD model showing the assembly and individual parts.

**Figure 3 sensors-23-02460-f003:**
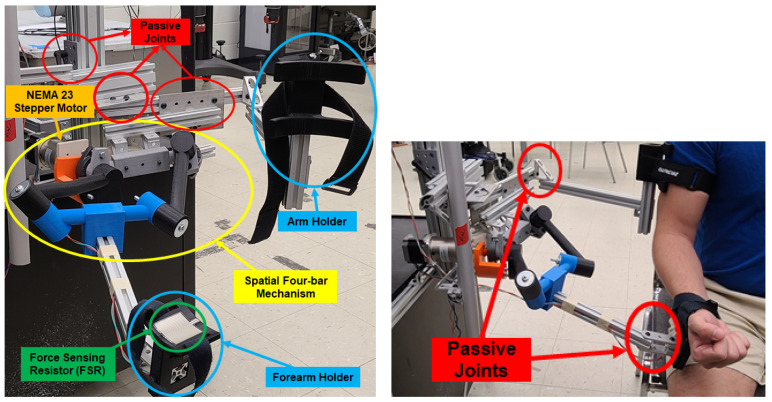
Task-based exoskeleton prototype.

**Figure 4 sensors-23-02460-f004:**
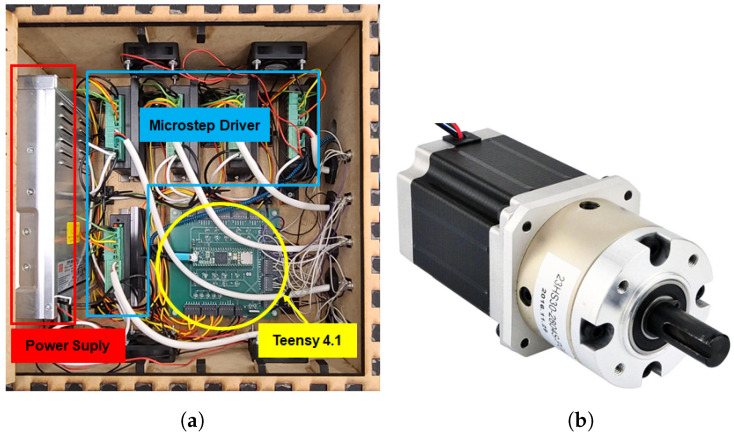
Control Box and NEMA 23 Stepper Motor. (**a**) Control Box. (**b**) NEMA 23 Stepper Motor with 46:1 Planetary Gearbox.

**Figure 5 sensors-23-02460-f005:**
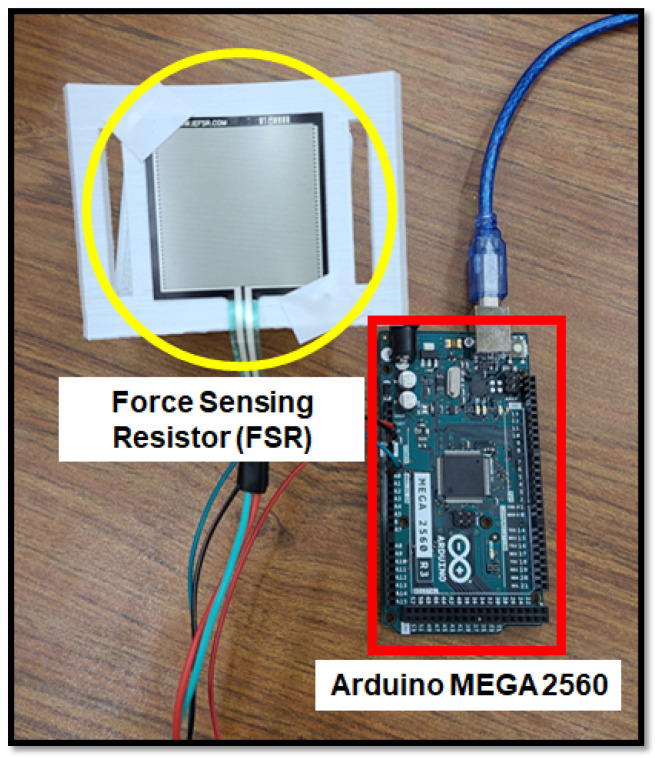
FSR sensor and Arduino setup.

**Figure 6 sensors-23-02460-f006:**
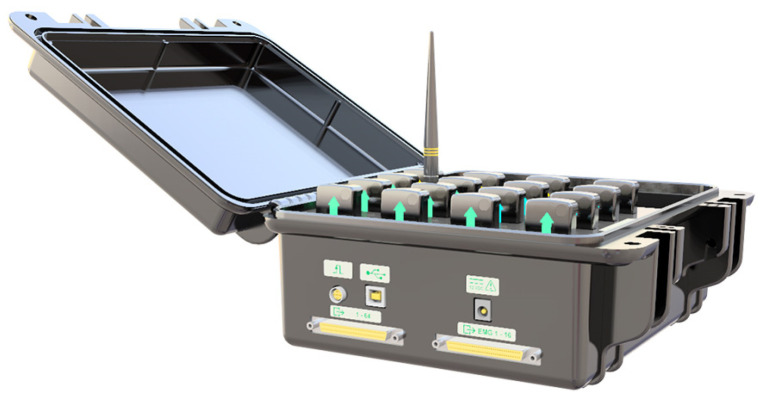
Delsys Trigno system.

**Figure 7 sensors-23-02460-f007:**
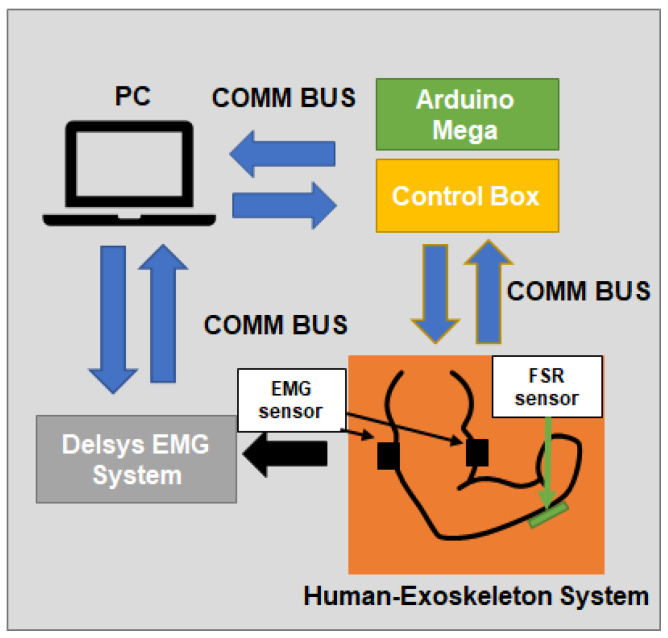
Interface schematic.

**Figure 8 sensors-23-02460-f008:**
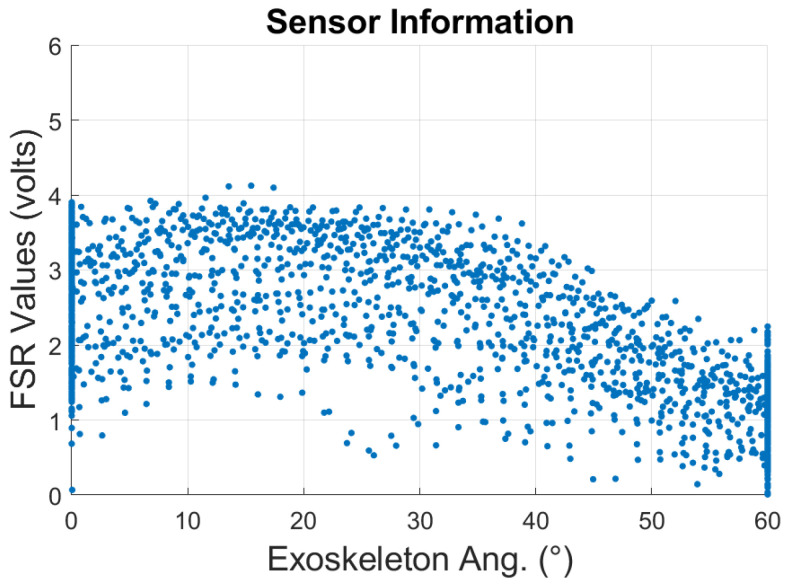
Scatter plot of the exoskeleton input values and FSR values for a subject during elbow flexion–extension multiple cycles.

**Figure 9 sensors-23-02460-f009:**
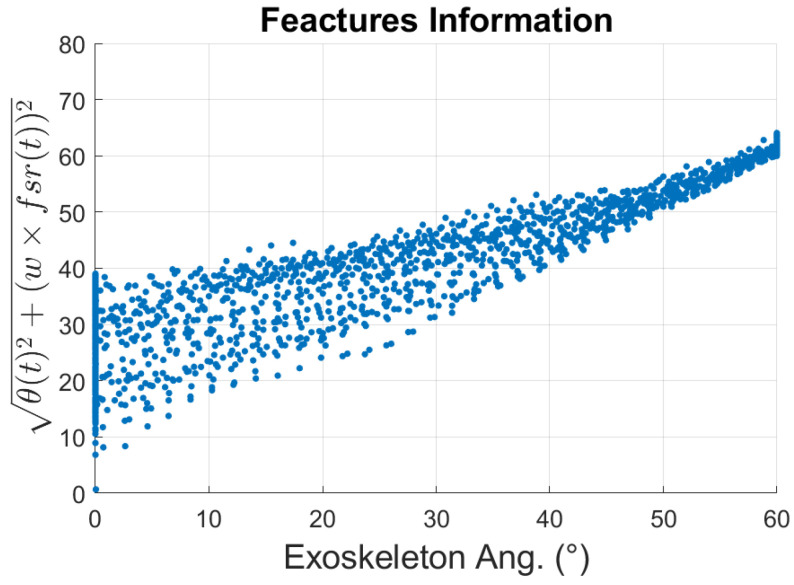
Scatter plot of the weighted *L*^2^-*norm* as the exoskeleton angles changed during elbow flexion–extension multiple cycles.

**Figure 10 sensors-23-02460-f010:**
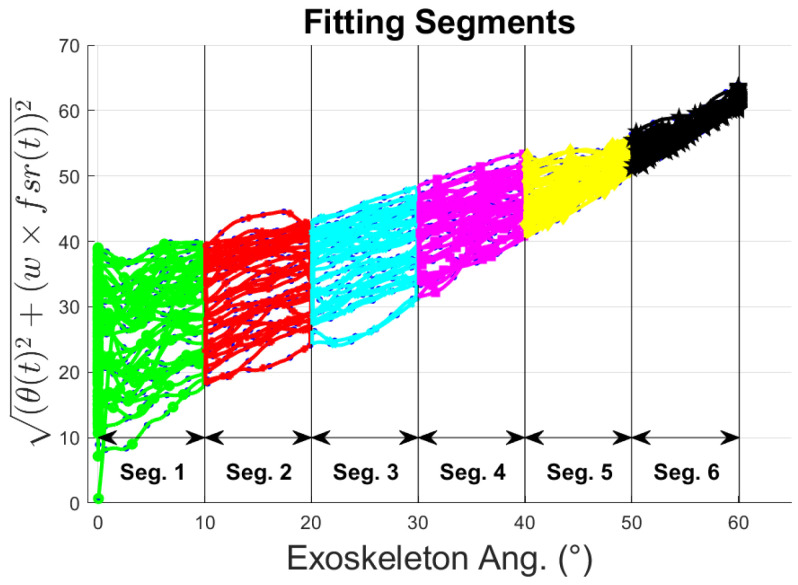
Segments to perform curve fitting model analysis.

**Figure 11 sensors-23-02460-f011:**
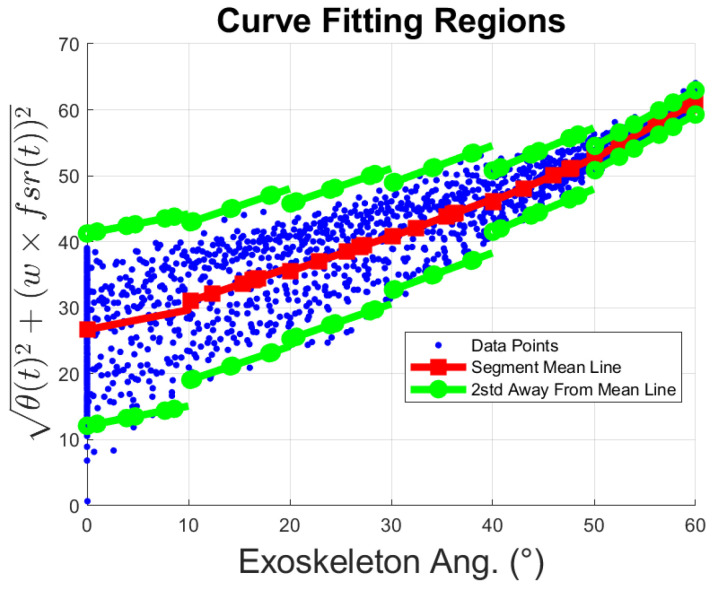
1st-order polynomial curve fitting by segments.

**Figure 12 sensors-23-02460-f012:**
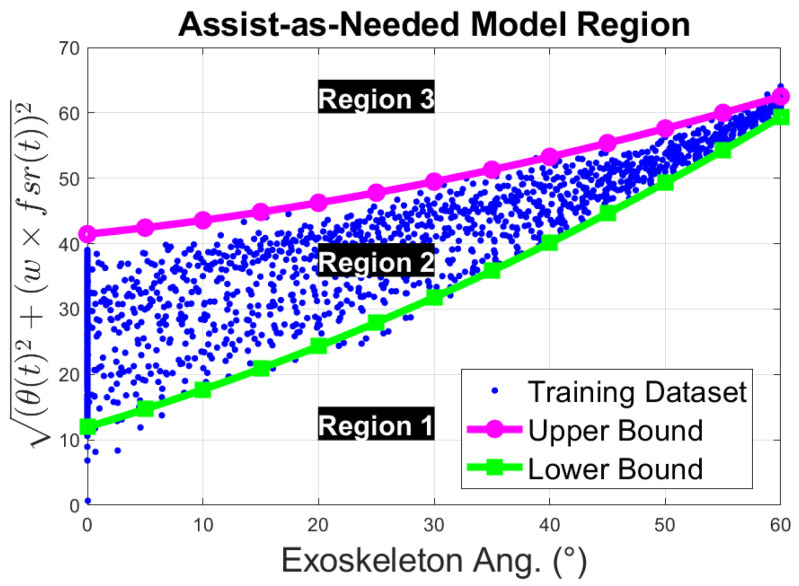
Assist-as-Needed trained model regions to infer subject’s action.

**Figure 13 sensors-23-02460-f013:**
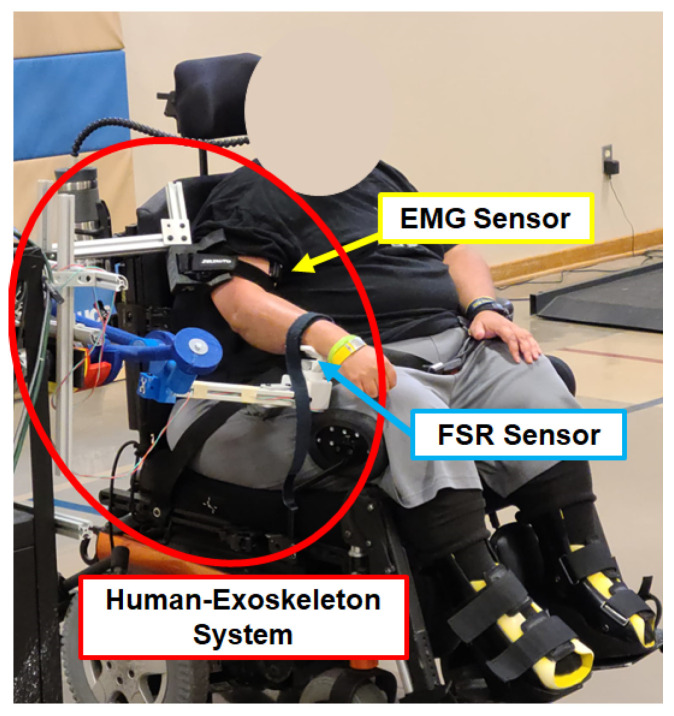
Human-exoskeleton system setup.

**Figure 14 sensors-23-02460-f014:**
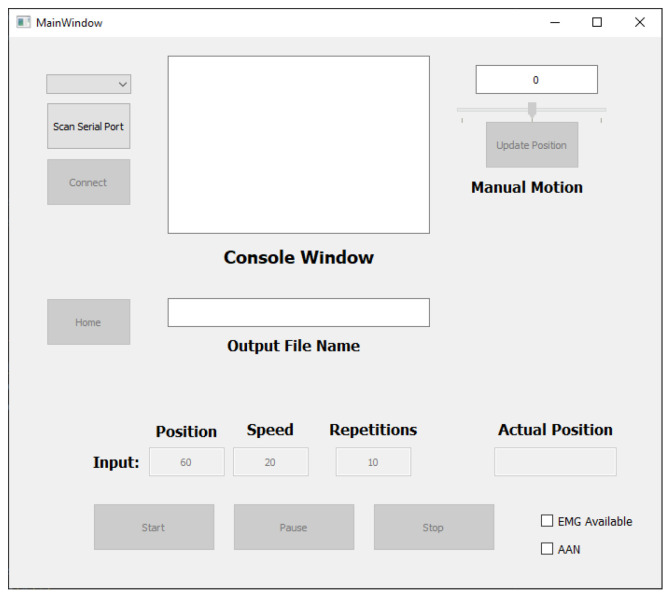
Python graphical user interface (GUI) to communicate with the exoskeleton.

**Figure 15 sensors-23-02460-f015:**
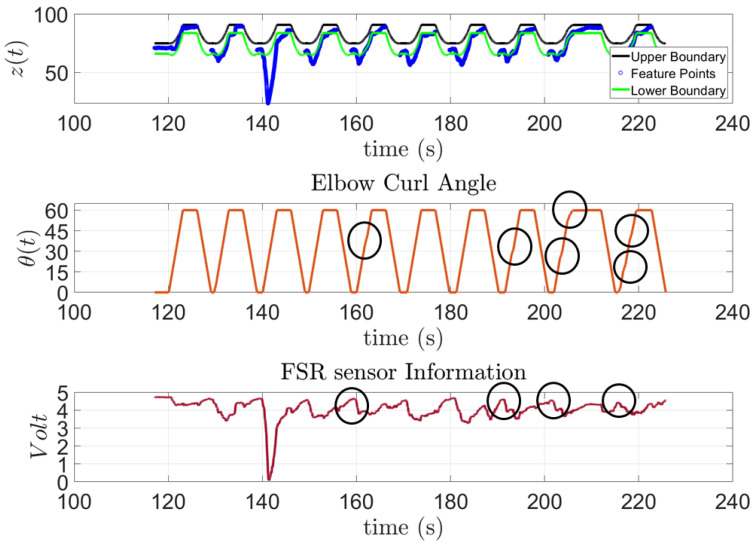
Subject 1: AAN through elbow flexion exercise.

**Figure 16 sensors-23-02460-f016:**
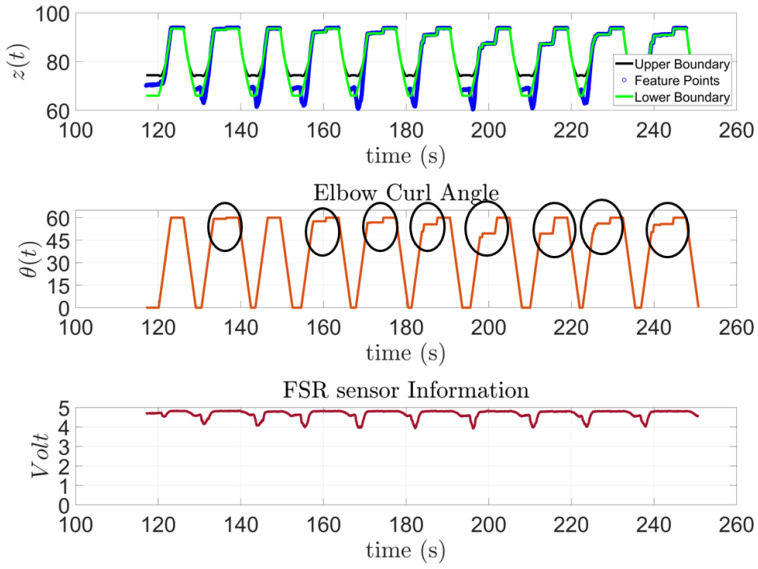
Subject 2: AAN through elbow flexion exercise.

**Figure 17 sensors-23-02460-f017:**
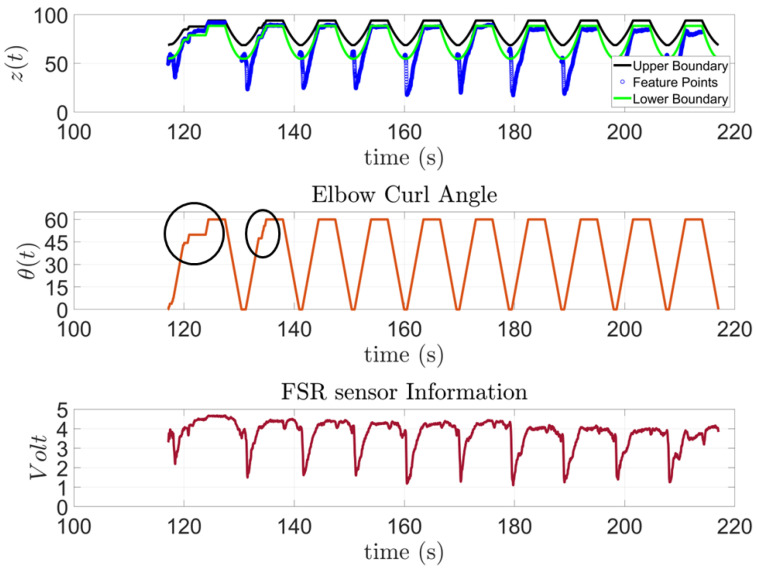
Subject 3: AAN through elbow flexion exercise.

**Figure 18 sensors-23-02460-f018:**
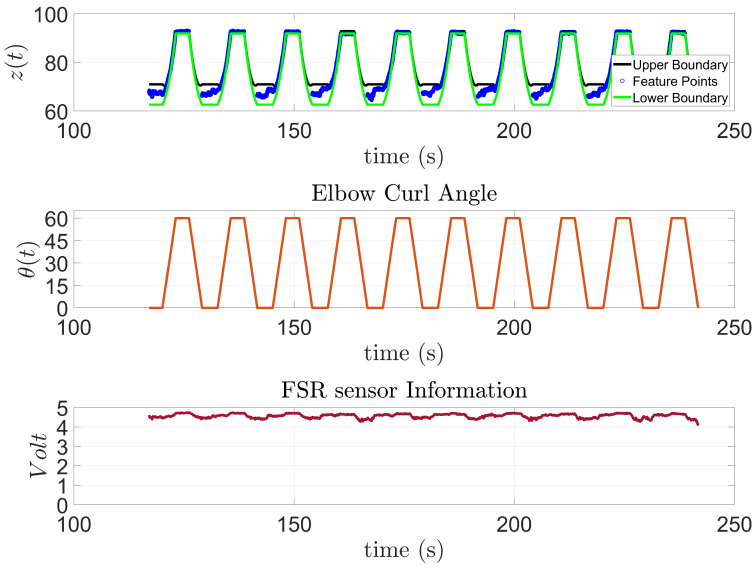
Subject 4: AAN through elbow flexion exercise.

**Figure 19 sensors-23-02460-f019:**
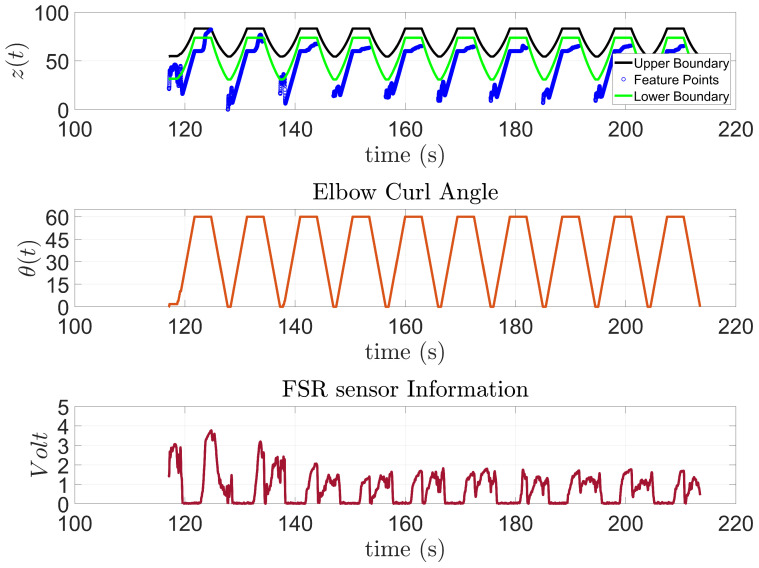
Subject 5: AAN through elbow flexion exercise.

**Figure 20 sensors-23-02460-f020:**
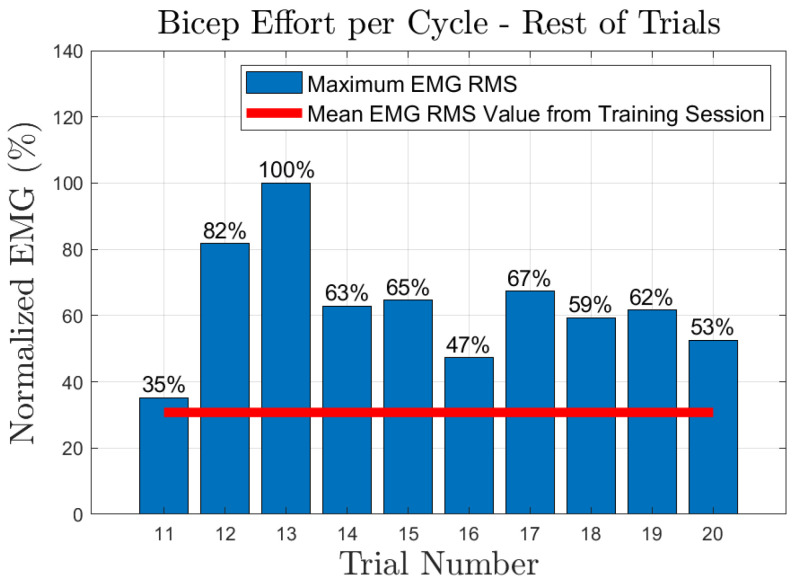
Subject 1: Flexion sEMG RMS values per trial.

**Figure 21 sensors-23-02460-f021:**
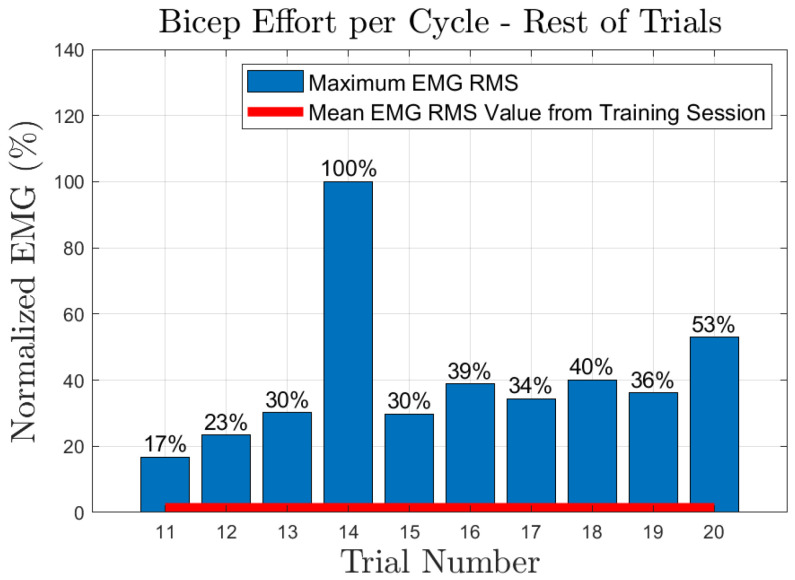
Subject 2: Flexion sEMG RMS values per trial.

**Figure 22 sensors-23-02460-f022:**
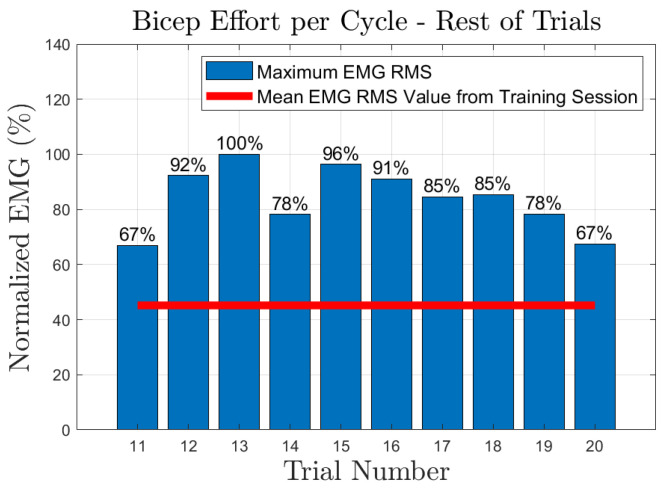
Subject 3: Flexion sEMG RMS values per trial.

**Figure 23 sensors-23-02460-f023:**
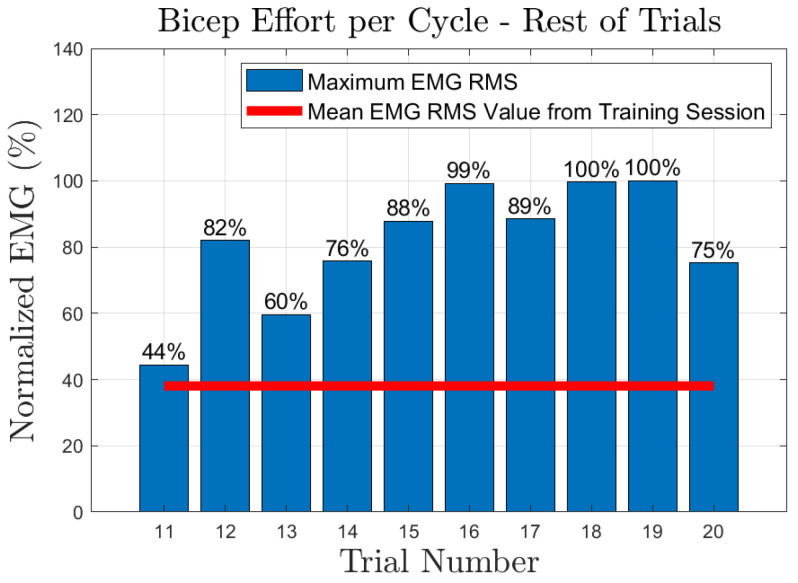
Subject 4: Flexion sEMG RMS values per trial.

**Figure 24 sensors-23-02460-f024:**
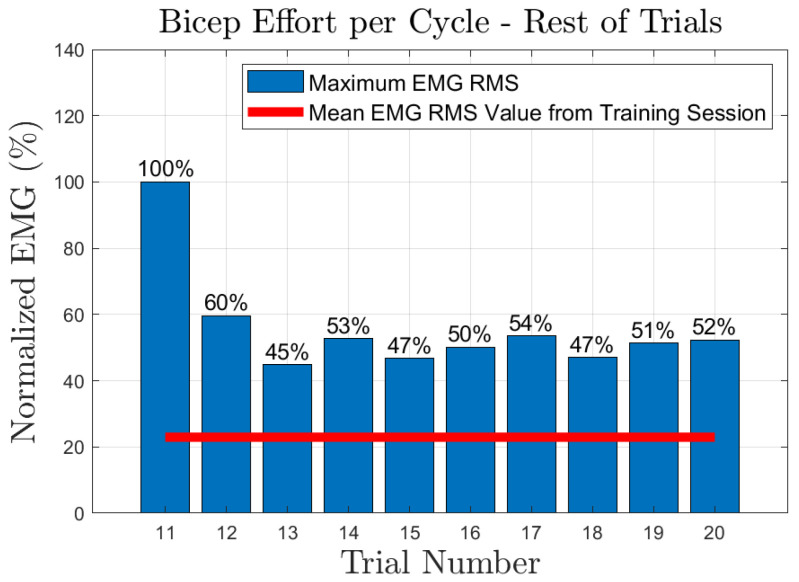
Subject 5: Flexion sEMG RMS values per trial.

**Table 1 sensors-23-02460-t001:** Model accuracy for each individual subject.

Subject	1	2	3	4	5
Model Accuracy	96.85%	77.91%	94.61%	93.82%	92.91%

## Data Availability

Data are available upon request.

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
