# Peer review of "Integration of Task-Based Exoskeleton with an Assist-as-Needed Algorithm for Patient-Centered Elbow Rehabilitation"

_sensors, 2023, doi:10.3390/s23052460_

Round 1
Reviewer 1 Report
1. Change Section 2 to become " Problem formulation and assumptions".
2. In the results, compare your results with pireviously published work.
Reviewer 2 Report
The paper proposes a control algorithm for automated elbow rehabilitation. A Bennet linkage is used to replicate the elbow's flexion-extension motion. This mechanism is shown to synthesize elbow kinematics better than a revolute joint, and it is replicated with a rigid-link design. After presenting the design, they show the acquired force through the onboard FSR sensor and discuss how to generate an assistance strategy from the results. They then report a case study with 5 injured patients.
The paper is interesting, but I would suggest the authors to address the comments below prior publication.
COMMENTS:
1) Why do the authors use a 1-DoF rigid-link mechanism? Especially as motion is shown as guided through compliant straps, the proposed rehabilitation device is not going to guide the patient though the exact trajectory. The proposed design should be also better motivated with respect to cable driven (e.g., [A-D]) and soft (e.g., [E, F]) designs.
[A] Xiong, Hao, and Xiumin Diao. "A review of cable-driven rehabilitation devices." Disability and Rehabilitation: Assistive Technology 15.8 (2020): 885-897.
[B] Zanotto, Damiano, et al. "Sophia-3: A semiadaptive cable-driven rehabilitation device with a tilting working plane." IEEE Transactions on Robotics 30.4 (2014): 974-979.
[C] Rodríguez-León, Jhon F., et al. "An Autotuning Cable-Driven Device for Home Rehabilitation." Journal of healthcare engineering 2021 (2021).
[D] Shoaib, M., Asadi, E., Cheong, J., & Bab-Hadiashar, A. (2021). Cable driven rehabilitation robots: Comparison of applications and control strategies. IEEE Access, 9, 110396-110420.
[E] Oguntosin, Victoria, et al. "Development of a wearable assistive soft robotic device for elbow rehabilitation." 2015 IEEE International Conference on Rehabilitation Robotics (ICORR). IEEE, 2015.
[F] Wu, Qingcong, et al. "Design and fuzzy sliding mode admittance control of a soft wearable exoskeleton for elbow rehabilitation." IEEE Access 6 (2018): 60249-60263.
2) How is the FSR sensor setup on the patient? I'd suggest the author show the exact placement during the experiment, as the results and their interpretation are strictly dependent on it.
3) It would be interesting to see the result of the same experiments reported in Section 5 as performed by healthy subject, to have an idea of "how much" the proposed strategy supports injured patients.
4) How is the model accuracy in Table 1 defined and computed? What does it represent, exactly? It is only mentioned in the table, but a formal (mathematica) definition of it is never reported, nor is a description.
5) For the results in Fig. 15 to 19, would it be possible to report also the measured force and assistive action corresponding to the motion?
Reviewer 3 Report
This paper proposes an assist-as-needed algorithm to provide real-timely visual feedback for the patients to complete the therapy sessions. Tests indicate that an accuracy of 91.22% can be achieved, which is a satisfying result. However, the paper needs minor revisions before publication. My questions are as follows:
1. As per your description in line 180, I cannot understand how a weighted L2 is determined and why ROM of elbow is set as 0-60 degrees.
2. If the participants cannot complete the elbow-flexion task, how to obtain the first 10 trials to train the proposed AAN algorithm? If the exoskeleton performs 10 elbow flexion and extension motions, what is the angular velocity?
3. Why are the difficulties observed at the beginning attributed to the subject (3 & 5) getting familiar with the exoskeleton in prediction trials? I think they have been very familiar with the usage of exoskeleton after the initial 10 elbow flexion and extension motions.
Reviewer 4 Report
This is exciting work on the assist-as-need (AAN) algorithm applied to a task-based elbow exoskeleton. The work is relevant, timely, and effective for rehabilitating the rear arm and forearm. The manuscript's organization is fair, and the results are well documented. Although the manuscript can be accepted in the present form, a few suggestions can be incorporated to improve the quality of the paper.
1. The novelty of the work could be explained in bullet/point form to improve the readability of the work.
2. An exploded view of the exoskeleton CAD could be presented in the manuscript.
3. The placement and orientation details of FSR and EMG sensors could be highlighted more.
4. The machine learning model used in the AAN algorithm should be specified if any, as mentioned in the abstract.
5. There should be a thorough language check for grammatical and punctuation errors.
Round 2
Reviewer 1 Report
The current version is OK.
Author Response
Thank you very much!
Reviewer 2 Report
The authors have addressed most of my concerns. However, their response to one of my comments missed the point. I'll report the comment here:
1) Why do the authors use a 1-DoF rigid-link mechanism? Especially as motion is shown as guided through compliant straps, the proposed rehabilitation device is not going to guide the patient though the exact trajectory. The proposed design should be also better motivated with respect to cable driven (e.g., [A-D]) and soft (e.g., [E, F]) designs.
The revision was a paragraph in the experiments section: "Once the patient is attached to the mechanism, they are physically constrained to follow the spatial displacement of the Bennet linkage without being affected or driven by the straps."
However, this does not address the fact that compliance in the strap affects the motion actually transmitted to the patient. Due to the strap's deformation and the skin's elasticity, what is transmitted to the patient is not the exact motion at the Bennet's linkage. Further, they did not compare this choice nor discuss it with respect of the literature (in which this issue is also often addressed).
